# Direct Visualization of Horizontal Gene Transfer by Transformation in Live Pneumococcal Cells Using Microfluidics

**DOI:** 10.3390/genes11060675

**Published:** 2020-06-20

**Authors:** Isabelle Mortier-Barrière, Patrice Polard, Nathalie Campo

**Affiliations:** Laboratoire de Microbiologie et Génétique Moléculaires, Centre de Biologie Intégrative (CBI), Centre National de la Recherche Scientifique (CNRS), Université de Toulouse, UPS, F-31000 Toulouse, France; isabelle.mortier@ibcg.biotoul.fr (I.M.-B.); patrice.polard@ibcg.biotoul.fr (P.P.)

**Keywords:** *Streptococcus pneumoniae*, competence, transformation, fluorescence microscopy, microfluidic

## Abstract

Natural genetic transformation is a programmed mechanism of horizontal gene transfer in bacteria. It requires the development of competence, a specialized physiological state during which proteins involved in DNA uptake and chromosomal integration are produced. In *Streptococcus pneumoniae*, competence is transient. It is controlled by a secreted peptide pheromone, the competence-stimulating peptide (CSP) that triggers the sequential transcription of two sets of genes termed early and late competence genes, respectively. Here, we used a microfluidic system with fluorescence microscopy to monitor pneumococcal competence development and transformation, in live cells at the single cell level. We present the conditions to grow this microaerophilic bacterium under continuous flow, with a similar doubling time as in batch liquid culture. We show that perfusion of CSP in the microfluidic chamber results in the same reduction of the growth rate of individual cells as observed in competent pneumococcal cultures. We also describe newly designed fluorescent reporters to distinguish the expression of competence genes with temporally distinct expression profiles. Finally, we exploit the microfluidic technology to inject both CSP and transforming DNA in the microfluidic channels and perform near real time-tracking of transformation in live cells. We show that this approach is well suited to investigating the onset of pneumococcal competence together with the appearance and the fate of transformants in individual cells.

## 1. Introduction

Natural genetic transformation is a programmed mechanism of horizontal gene transfer entirely dependent on proteins encoded by the recipient bacteria [1,2]. It involves the uptake of DNA from the environment and its integration into the bacterial chromosome by homologous recombination. Considerable knowledge has accumulated on this mechanism since its discovery in 1928 by Frederick Griffith in the human pathogen *Streptococcus pneumoniae* (the pneumococcus) [1,2,3]. The number of species known to be naturally transformable is less than a hundred, but phylogenetic analysis indicated that transformability is spread throughout the main taxa, including both Gram-positive and Gram-negative bacteria [1]. In most transformable bacteria, the ability to transform is transient. It requires the development of the differentiated state of competence, during which cells can import DNA for genetic diversity or repair their chromosome [1,2]. As such, in *S. pneumoniae*, natural transformation allows acquisition of pathogenicity islands, antibiotic resistance and promotes vaccine escape via capsule switching [4].

A unique feature of pneumococcal competence is its populational character, as all cells develop competence for about 25 min at the onset of the exponential phase of planktonic growth. This coordination of competence in the growing cell population relies on a secreted peptide pheromone, the Competence-Stimulating Peptide (CSP) that spreads through the population by cell–cell contact [5]. Importantly, synthetic CSP externally added into pneumococcal cultures efficiently induces competence in a broad range of conditions and culture media [6]. It triggers the transcriptional activation of a global regulon comprising approximately 100 genes [7,8,9,10]. This regulon is made of two classes of genes with distinct temporal expression profiles defined as ‘early’ and ‘late’ com genes. The early class includes genes necessary for the activation of the competence regulatory cascade, while the genes required for the uptake of DNA and its processing into recombinants are part of the late class. The kinetics of recombinant formation indicate that integration of internalized DNA into the recipient chromosome occurs in less than 15 min [11,12]. Integration can occur into either strand of the recipient DNA with equal probability [13,14,15]. The resulting heteroduplex is subsequently resolved by replication and segregation, producing one daughter chromosome identical to the recipient genome and one containing the donor sequence. The time needed for the phenotypic expression of the donor DNA is thus impacted by whether it was integrated on the coding or non-coding strand.

Most of the information acquired on pneumococcal competence and transformation comes from population data with approaches ranging from classical molecular genetics and biochemistry, to modern high throughput techniques based on genomics, transcriptomics and proteomics. These methods are powerful to analyze the average behavior of competent cells, especially given that all pneumococcal cells are able to develop competence in a coordinated manner. More recently, single cell analyses using fluorescence microscopy have been used to detect competent cells [16,17,18,19,20], to localize transformation proteins [12,15,21], to analyze the binding of transforming DNA at the surface of competent cells and to visualize transformants [12,15]. The development of the droplet microfluidic technology to study transformation also allowed the characterization of the genetic outcome of individual cell–cell interactions. However, heterogeneity in the population and more specifically, the tracking of an individual cell developing competence and undergoing the process of transformation are still poorly studied. 

Here, we present a method to visualize the entire transformation process in live pneumococcal cells at the single cell level. Our strategy is based on a combination of microfluidics and time-lapse fluorescence microscopy. It provides a unique opportunity to monitor the onset of competence induction and to follow the different steps leading to the phenotypic expression of pneumococcal transformants.

## 2. Materials and Methods 

### 2.1. Bacterial Strains, Culture and Transformation Conditions

*S. pneumoniae* strains, plasmids and primers used in this study are described in Appendix A. *S. pneumoniae* strains were all constructed in the R1501 background, which is derived from strain R800 [22]. This strain contains the *ΔcomC* mutation and cannot develop competence spontaneously [7]. Stock cultures were routinely grown at 37 °C to OD550~0.3 in Todd-Hewitt medium (BD Diagnostic System, Sparks, MD, USA) supplemented with 0.5% Yeast Extract (THY) or C+Y medium [23]; after addition of 15% (vol/vol) glycerol, stocks were kept frozen at −70 °C. For the monitoring of growth, pre-cultures grown in C+Y medium to OD550~0.3 were inoculated (1 in 50) in C+Y medium complemented or not with catalase (300 U/mL) and distributed into a 96-well microplate (300 µL per well). OD values were recorded throughout incubation at 37 °C in a Varioskan luminometer (ThermoFisher Scientific Oy, Vantaa, Finland). For the study of competence development, cells were incubated in C+Y medium with synthetic CSP.

CSP-induced transformation was performed as described previously [24]. Measurement of transformation efficiency was carried out as described [18,19], with R304 chromosomal DNA carrying the *rpsL41* point mutation conferring resistance to streptomycin (SmR). Antibiotic used for the selection of *S. pneumoniae* transformants were: kanamycin for strain constructions (250 µg/mL, and streptomycin for transformation assays (200 µg/mL). 

### 2.2. Strain Constructions

Strains harboring transcriptional fusions of the *gfp* (strain R4254), *mTurquoise2* (strain R4255) and *mCherry* (strain R4256) reporter genes under the control of the promoter of the early competence operon *comCDE*, P_E_, were obtained by transformation of strain R1502 with plasmid pIM122 and strain R1501 with plasmids pIM123 and pIM124 respectively. These plasmids are integrative plasmids derived from pCEP_E_ [25]. pCEP_E_ allows chromosomal integration of a gene at CEP (Chromosomal Expression Platform) and its expression under the control of the CSP-inducible, ComE-dependent promoter P_E_. To generate pIM122, the *gfp*(*Sp*) gene with codons optimized for *S. pneumoniae* was amplified with the oMB2 and oCN87 primer pair using the pUC57-*gfp*(*Sp*) plasmid as template [16]. To create pIM123, the *mTurquoise2*(*Sp*) gene with codons optimized for *S. pneumoniae* was amplified with the oIM78 and oIM79 primer pair using the pUC57*-mTurquoise2*(*Sp)* plasmid as template [23]. To obtain pIM124, the gene encoding mCherry was first synthesized with codons optimized for *S. pneumoniae* strain R6 (http://gib.genes.nig.ac.jp/) and cloned into pUC57 by Genscript USA to generate plasmid pUC57-*mCherry*(*Sp*). The *mCherry* gene was then amplified with the oIM80 and oIM81 primer pair using pUC57*-mCherry*(*Sp)* plasmid as template. All PCR products were subsequently cut with *Nco*I and *Bam*HI, and inserted into pCEP_E_ between *Nco*I and *Bam*HI to generate plasmids pIM122 (i.e., pCEP_E_-*gfp*(*Sp*)), pIM123 (i.e., pCEP_E_-*mTurquoise2*(*Sp*)) and pIM124 (i.e., pCEP_E_-*mCherry*(*Sp*)).

### 2.3. Time-Lapse Microfluidic Microscopy

Microfluidic experiments were performed using the CellASIC^®^ ONIX Microfluidic Platform and B04A microfluidic plates for bacterial cells (Merck-Millipore, Billerica, MA, USA). To completely remove PBS, microfluidic chambers and loading channels were first washed with C+Y medium complemented or not with catalase (300 U/mL) with 5 psi for 5 min at 37 °C, and further incubated for 30 min under continuous pressure (0.25 psi). Note that according to the indications of the manufacturer, a pressure of 0.25 psi corresponds to a theoretical flow rate of 0.3 µL/h. Moreover, given the dimensions of the CellASIC^®^ ONIX microfluidic chamber, the time necessary to completely renew the solutions in the chamber should be 52 s at a flow rate of 0.3 µL/h (0.25 psi/1.72 kPa), 1.6 s at 10 µL/h (3 psi/20.68 kPa) and 0.7 s at 23 µL/h (6 psi/41.37 kPa).

Exponential growing cultures (OD550 0.3) were diluted 50-fold in C+Y medium complemented or not with catalase (300 U/mL) and incubated at 37 °C to an OD550 of 0.1 (~10^8^ cells/mL). Cells were subsequently loaded into the microfluidic chamber according to manufacturer’s protocol. Note however that the concentration of the cell suspension used for loading was about 10 times higher than the concentration recommended by the manufacturer, which is 1 to 20 × 10^6^ cells/mL. We found that pneumococcal cells were preferentially immobilized into the fifth trap (trap height of 0.7 µm). Cells were routinely maintained at 37°C in a thermostated chamber with a constant flow rate of 0.3 µL/h (0.25 psi) in C+Y medium supplemented with catalase (all figures, except Figure 1 and Appendix A). Competence induction was achieved by injecting CSP diluted at a concentration of 500 ng/µL in C+Y medium and catalase under pressure (2 min at 6 psi, followed by 6 min at 3 psi).

Images were captured and processed using the Nis-Elements AR software (Nikon Instruments Europe BV, Amsterdam, The Netherlands). Phase contrast and fluorescence microscopy were performed with an automated inverted epifluorescence microscope Nikon Ti-E/B equipped with the “perfect focus system” (PFS, Nikon), a phase contrast objective (CFI Plan Apo Lambda DM 100X, NA1.45), Semrock filters sets for Green Fluorescent Protein (GFP) (Ex: 482BP35; DM: 506; Em: 536BP40), CFP (Ex: 438BP24; DM: 458; Em: 483BP32) and mCherry (Ex: 562BP40; DM: 593; Em: 641BP75), a LED light source (Spectra X Light Engine, Lumencor, Beaverton, OR, USA), and sCMOS camera (Neo sCMOS, Andor, Belfast, UK), and a thermostated chamber at 37 °C. All fluorescence images were acquired with a minimal exposure time to minimize bleaching and phototoxicity effects. Acquisition settings were 500 ms for the transcriptional fusion to GFP using 50% power of a LED light source at 470 nm, 800 ms for the transcriptional fusion to mCherry and 600 ms for the transcriptional fusion to mTurquoise, using 20% power of a LED light source at 555 nm and 440 nm excitation wavelengths, respectively. GFP, mCherry and mTurquoise fluorescence images were respectively false colored green, red and blue, and overlaid on phase contrast images.

### 2.4. Image Analysis

The quantification of the fluorescence was performed as follows. The outlines of single cells were first detected using the phase contrast images and the threshold command from Nis-Elements. The fluorescence levels were corrected for background fluorescence. For this, we used wild-type cells as negative controls to monitor autofluorescence and set detection thresholds. The object measurement tool was used to obtain the total fluorescence and the total volume of the cells for each image, in order to calculate the average fluorescence intensity per µm^2^. This procedure was repeated for each time point and the average fluorescence intensity (in arbitrary unit) was plotted as a function of time. Note that although more than 200 cells were counted for each replicate, a strict segmentation of single cells was not necessary for this analysis.

To measure doubling times, we manually tracked single cell lineages up to seven generations using phase-contrast time-lapse microscopy images. Alternatively, we used overlaid phase contrast and fluorescence images of cells harboring a functional FtsZ-GFP fusion as a marker for cell division, as previously described [21]. A minimum of 50 cell lineages were analyzed per experiment and the experiments were repeated 2 to 4 times.

### 2.5. Direct Visualization of Transformation Assay

The protocol described previously to track the formation of transformants in live cells [12], was adapted to visualize all steps from the onset of competence development. We used strain R3708 harboring the *gfp* coding sequence joined in frame to the 3′ end of *ftsZ* but separated from it by a stop codon (TAA) as a recipient strain [12]. Cells cultivated in C+Y medium supplemented with catalase were loaded into the CellASIC^®^ microfluidic chamber and incubated under a constant flow rate of 0.3 µL/h (0.25 psi), for 30 min. Competence was induced by exchanging the medium in the chamber with C+Y medium containing CSP (see above), immediately followed by perfusion of C+Y medium containing donor DNA (30 ng/µL) during 5 min at 10 µL/h (3 psi). Donor DNA consisted in a 2.7-kb PCR fragment encoding the functional *ftsZ-gfp* fusion amplified from strain R3702 chromosomal DNA using the OMB94-OMB97 primer pair. Transformed cells were detected by monitoring FtsZ-GFP expression. R4256 cells containing the P_E_-*mCherry* transcriptional fusion were used as a control to verify effective development of competence in the microfluidic chamber. Cells were imaged every 4 min for 2.5 h. Acquisition settings were 400 ms for GFP using 50% power of a LED light source at 470 nm excitation wavelengths and 800 ms for mCherry using 20% power of a LED light source at 555 nm. Note that mCherry images were acquired every 10th time point.

## 3. Results

### 3.1. Optimisation of Pneumococcal Growth in Microfluidic Chambers

To maintain the growth of pneumococcal strains overtime, and to observe the induction of competence and the transformation process in real time, we used the CellASIC^®^ ONIX microfluidic platform (see Section 2.3). Our first attempts at cultivating cells under a constant flow rates of 0.3 µL/h (0.25 psi) or 2.5 µL/h (1 psi) were unsuccessful. These flow rates were sufficient to renew the growth medium inside the microfluidic chamber every 52 and 6 s, respectively. However, the generation time gradually increased and cells stopped dividing after a few divisions (Figure 1). Fluidic flows within microsystems are known to generate shear stress and to raise the concentration of dissolved oxygen [26]. *S. pneumoniae* is a microaerophile bacterium, which can grow in the presence of low levels oxygen. Under aerobic conditions, it produces high concentrations of hydrogen peroxide (H_2_O_2_) and necessitates the addition of scavengers to the culture for optimal growth rate [27]. We considered that, while a higher flow rate may increase oxygen in the microfluidic chamber, it could also help wash out or reduce the amount of H_2_O_2_ produced. Consistent with this hypothesis, an augmentation of the flow rate to 6 µL/h (2 psi) allowed cells to proliferate for at least 7 to 8 generations (Appendix A). Similarly, quenching H_2_O_2_ by adding catalase in the medium promoted cell growth up to confluence with a steady doubling time of 24 min (Figure 1, see Section 2.1 and Section 2.3). Notably, the presence of catalase enhanced growth in batch liquid culture and lessened cell lysis (Appendix A, and as previously shown [27]), but had no effect on the transformation efficiency (Appendix A). Based on these results, and with the objective of limiting shear stress, we performed all subsequent experiments with slow flow rates (0.3 µL/h) and catalase (300 U/mL).

### 3.2. Robust Competence Development in Microfluidic Chambers

To monitor the development of competence in individual cells, we used a strain expressing the GFP under the control of the promoter of the early *comCDE* operon (P_E_, strain R4254). For this, we directly recorded the fluorescent signal in real time upon perfusion of growth medium supplemented with CSP in the microfluidic chamber. Synthetic CSP, which is a small 17 amino-acid peptide containing hydrophobic patches [28,29], tends to be absorbed into the polydimethylsiloxane (PDMS) matrix composing the microchannels [30]. A potential solution to circumvent this difficulty and ensure medium exchange was to perfuse CSP at high flow rate for a short time period. Nevertheless, we found that increasing the flow rate more than a 75-fold (23 µL/h, 6 psi) was not sufficient to obtain rapid and efficient competence induction at CSP concentrations used in batch cultures (125 ng/µL, Appendix A). Indeed, although fluorescence was detected in most of the cells, the signal intensity remained barely above background for the first 10 to 15 min and did not reach the highest expected levels, even after 30 min. However, increasing the concentration of CSP 4-fold allowed full competence induction. Within 10 min, all cells showed high fluorescence levels that continued to accumulate during the next 20 min. The duration of CSP perfusion also impacted the detection time and the intensity of the fluorescent signal (Figure 2a). Altogether, optimal conditions to achieve rapid and efficient competence induction involved perfusing CSP at a concentration of 500 ng/µL for 8 min in two sequential steps. CSP was first added for 2 min at a high flow rate of 23 µL/h to initiate medium exchange, followed by 6 min at 10 µL/h before switching back to standard growth medium (without CSP) and flow rate (0.3 µL/h).

To further test the robustness of these perfusion parameters for competence development, we analyzed a visual morphological feature of pneumococcal competent cells. We have shown previously that competent cells undergo a CSP-dependent cell division delay [21]. Using a strain harboring an FtsZ-GFP fusion (strain R3702) as a marker for cell division in the microfluidic device, we confirmed that CSP perfusion resulted in a reduction in the growth rate of individual cells (Figure 2B). For this experiment, cells were loaded in two parallel chambers of the microfluidic device. Growth medium containing CSP was flushed in one chamber according to the protocol indicated above. At the same time, and with similar flow rates, medium without CSP was introduced into the second chamber. Measuring the doubling time of individual cells from the beginning of the perfusion demonstrated that CSP instantly causes a delay of the division process (Figure 2B,C). Non-competent cells maintained a doubling time of 28.4 ± 3.5 min, even at higher flow rates. In contrast, the doubling time of competent cells increased to 40.2 ± 5.6 min, in line with the values reported in previous work [21]. Importantly, this analysis also verified that daughter cells derived from CSP-treated cells immediately recovered doubling times similar to non-competent cells (Figure 2C, [21]). Altogether, these results validate the use of the microfluidic technology to study pneumococcal competence development in time and at the single cell level.

### 3.3. Temporal Visualization of Early and Late Com Genes Expression

With the aim of studying the temporal expression of different competence genes, we developed several fluorescent reporter strains. We first choose three spectrally distinct fluorescent proteins with minimal or no cross talk, namely, GFP, the blue fluorescent protein mTurquoise and the red mCherry protein. To test the suitability of these proteins for simultaneous visualization in pneumococcal cells, we constructed transcriptional fusions between the early competence promoter P_E_ and the coding sequences of the corresponding genes. Notably, all these sequences were codon optimized for expression in *S. pneumoniae*. To compare the kinetics of expression of the reporter genes, each strain was incubated in distinct chambers of the microfluidic device and simultaneously induced to develop competence upon CSP perfusion. Figure 3 shows that the three reporters were readily expressed and emitted fluorescence. Importantly, the analysis of the whole fields of view of the camera indicated that all cells develop competence, which suggests that CSP diffused throughout the entire microfluidic chambers (Appendix A). The fluorescent signal was, however, not detected at the same time after CSP perfusion in the three strains (Figure 3A and Appendix A). This difference can be explained by the inherent properties of each fluorophore. Indeed, the brightness of a fluorophore is defined by two parameters, the extinction coefficient (quantity of light absorbed) and the quantum yield (number of emitted photons per photon absorbed). It is also known that fluorescent proteins must undergo a maturation step to become fluorescent and that maturation times are highly variable for different fluorescent proteins [31]. In agreement with the published values of these parameters [31,32], the GFP signal was first detected in our experiments, followed by mTurquoise and then mCherry (Figure 3 and Appendix A). Interestingly, we found that the mTurquoise and mCherry fluorescence accumulated in daughter cells, suggesting that proteins synthesized during competence continued to mature and became visible well after competence shutoff (Appendix A and [25]). Altogether, these results demonstrate that the fluorescent proteins tested in this study cannot be used in combination to compare the temporal expression of different genes.

An alternative to overcome the inherent difficulties of using a combination of fluorophores, was to generate distinct reporters based on the same fluorescent protein. To estimate the lag between the expression of early and late *com* genes at the single cell level, we used a *gfp* transcriptional fusion with the early *com* promoter P_E_ (strain R4254), and a *gfp* translational fusion with the gene encoding the late competence protein DprA (strain R3728, to be published elsewhere). These reporters can be easily distinguished according to their cellular localization. Free GFP driven by the P_E_ promoter produced a diffused cytoplasmic fluorescent signal while DprA-GFP concentrated into bright discrete foci in competent cells (Figure 4 and Appendix A). Strains containing these constructions were loaded into the same microfluidic chambers and induced to develop competence upon CSP perfusion. Note that we did not introduce the two reporters in the same strain to avoid potential difficulties to detect the onset of DprA-GFP synthesis in cells expressing diffused GFP. In accordance with previous work indicating that late *com* genes expression is delayed by several minutes compared to early genes [7,10,33], we measured an interval of 4 to 5 min between the detection of the GFP signal in the cytoplasm and the formation of foci. This interval appeared slightly longer than the lag of 2 min formerly determined using transcriptional fusions with the Firefly luciferase gene [33]. We attribute this difference to the influence of the translational fusion with DprA that perturbs the folding of GFP, and therefore increases the maturation time compared to free GFP produced from a transcriptional fusion, and/or to the time needed to generate these foci. We reasoned however that the lag between the expression of early and late *com* genes was in good agreement with previously reported data [7,33]. We conclude that this method can be used as a readout to study the expression of early and late *com* genes at the single cell level.

### 3.4. Near Real-Time Visualization of Pneumococcal Transformation

We previously developed an assay to observe the transformation process by fluorescence microscopy in living cells grown on an agarose pad [12]. This assay consists of the transformation of a recipient cell (strain R3708, *ftsZ*-stop-*gfp*), with a ‘*ftsZ-gfp*’ donor DNA fragment carrying a mismatch of 3 bp and allowing expression of an FtsZ-GFP fusion upon integration into the chromosome (see Section 2.5). Combined with time-lapse fluorescence microscopy, it reports the appearance of transformants in near real time, by tracking the fluorescent FtsZ-GFP signal. These experiments, however, did not allow the observation of the development of competence. Indeed, competent cells were incubated with donor DNA in batch cultures for several minutes before spotting on an agarose pad and imaging [12]. We thus adapted this method with the microfluidic technology to be able to visualize the transformation process from the very onset of competence induction to the expression of newly acquired genes. To distinguish the timing of competence development from the timing of the expression of transformants, we used a mixed culture. This culture contained cells from strain R4256 (mCherry reporter strain for the development of competence) and strain R3708 (recipient strain for transformation with an *ftsZ-gfp* donor DNA fragment). This mixed culture was loaded into the microfluidic chamber and cultivated for one generation under a constant slow flow rate (0.3 µL/h). We then induced competence by injecting CSP for 8 min at high flow rate (see above and Section 2.3), subsequently followed by the addition of donor DNA for 5 min at 10 µL/h. It should be noted that in these conditions, we used a concentration of 30 ng/µL of the donor PCR DNA fragment harboring the *ftsZ-gfp* construct. This DNA concentration is 30-fold higher than the saturating amount of DNA that we use for optimal transformation assays in batch liquid culture with PCR DNA fragments carrying a single point mutation [12]. The requirement of a large amount of donor DNA to achieve transformation in the microfluidic device could be caused by several factors impacted by the flow rate, including a shorter contact time between the cells and the DNA and an increased shear stress on the DNA molecules.

Image analysis showed that cells expressing mCherry or the FtsZ-GFP fusion were homogeneously distributed in the microfluidic chamber (Appendix A). This result indicates that both CSP and DNA were capable of diffusing into the microfluidic device. To evaluate the proportion of transformed cells, we monitored, from the beginning of the DNA perfusion, the lineage of 107 individual cells that did not develop the mCherry fluorescence. We found that 29 of those cells generated transformants. Notably, all transformants produced two daughter cells with a bright FtsZ-GFP signal. The fluorescence intensity of these cells however, most often gradually disappeared in one of the siblings by dilution of the fusion protein over successive cell divisions (Figure 5, white arrows). This observation is most likely explained by an integration of the donor DNA into only one chromosomal strand. In that case, expression of the newly acquired gene allows the accumulation of the FtsZ-GFP fusion in the transformed cell and its transmission to the progeny by cell division. Yet, as resolution of the transformation heteroduplex by replication generates one transformed and one parental chromosome, only one daughter cell has the potential to give rise to a truly transformed population, as previously observed [15,34,35]. The first transformants exhibiting the FtsZ-GFP signal were detected before division of the transformed cell (17 cells, 59% of the transformed cells, Figure 5, white arrowhead). Concretely, the first transformants were observed 20 min after the addition of donor DNA. These results are consistent with an integration of the donor DNA into the non-coding strand of the recipient chromosome since this strand can produce messenger RNA before replication has occurred. We also detected 41 % of the cells expressing FtsZ-GFP only after division of the transformed cell (Figure 5, yellow arrowhead). Altogether, and in line with earlier reports [13,14,15], these findings indicate that either strand of the recipient chromosome can serve for transformation and that their transformability is statistically comparable.

## 4. Discussion

In this study, we used a microfluidic system and fluorescence microscopy to examine the growth and the transformation process of the human pathogen *S. pneumoniae* in near real-time and at the single cell level. This new approach represents a remarkable improvement to study the development of competence concomitantly with the appearance of transformants. It also offers the opportunity to reassess earlier findings based on genetic analyses or laborious and time-consuming methods involving radioactive substrates. Among these is the fate of the transformed cells. We confirm that transformed cells can express newly acquired genes before or after the first cell division event, probably depending on which strand of the recipient chromosome was transformed. We also directly visualized the phenomenon of non-genetic inheritance recently demonstrated in *Vibrio cholerae* and *S. pneumoniae* [15,36]. This mechanism, for which genetic evidence was presented in the 60s [34,35], involves the transient transmission of the transformed phenotype from the transformed cell to the part of its progeny that is not genetically transformed. The duration of the transmission in the progeny depends on the half-life and the amount of protein produced by the newly acquired gene before cell division of the transformed cell. In our experiments, we found that the FtsZ-GFP protein is transmitted over 2 generations.

Microfluidic systems, which allow the constant renewal of the culture medium and prevent accumulation of metabolic waste products, have proven to be useful to ensure steady state conditions for long-term observations of bacterial growth. Our results indicate that in the case of microaerophile bacteria such as the pneumococcus, catalase may be added to the culture media to ensure optimal growth in the microfluidic chamber. A major advantage of the microfluidic technology is also the possibility to visualize, in real time, how cells respond to the onset of a stress or any changes in their environments. Earlier reports, based on the measurement of the luciferase reporter enzyme activity, have shown that the time to initiate transcription of early *com* genes following the addition of saturating amounts of CSP in bulk cultures is about 2 min [33]. Similar measurements are not possible with cells grown on an agarose pad. Indeed, the very short time window necessary to activate *com* genes would make it technically challenging to observe the first cells entering competence without being able to introduce the CSP during the microscopy observation. Here, we overcame this difficulty but we found that the time required to detect the first cells expressing GFP under the control of an early competence promoter was more than 8 min (Figure 2 and Appendix A). The larger delay measured in our experiments can be explained by several factors including the time for the CSP to reach the microfluidic chamber and the maturation rate of the GFP fluorescent protein. In contrast, the interval between the expression of early and late *com* genes is consistent with values recorded by other methods, including RNA-seq analyses [9,10], and the use of luciferase or β-galactosidase reporters [7,8,33]. In addition, we found that the first transformants exhibiting an FtsZ-GFP signal appeared about 35 min after the onset of CSP perfusion. Considering that competence develops shortly after medium exchange with CSP, and knowing that DNA uptake is maximal 10 min after competence induction [12], the kinetics for the phenotypic expression of transformants are in line with results obtained from transformation assays [34]. Interestingly, however, and in agreement with the fact that microfluidic systems are better suited to continuous imaging than agarose pads, transformants were detected in the microfluidic chamber about 15 min earlier than in agarose pad-based experiments [12].

These findings imply that once activated with CSP, and in the presence of donor DNA, pneumococcal cells grown in microfluidic chambers undergo both the competence program and the transformation process with the same kinetics as in bulk cultures. The method presented here is therefore well suited to investigating the different steps of the transformation process in time, from the development of competence to the integration of donor DNA into the recipient chromosome and the phenotypic expression of the recombinants. The ability to perform these analyses at the single cell level also provides the possibility to study how the transformation mechanism is integrated into the cell cycle. This is particularly relevant as pneumococcal competence develops during exponential growth and is accompanied by a cell division delay (Figure 2 and [25]). Measuring the doubling time of individual cells over several generations confirmed that this arrest only takes place during the first division event following CSP addition (Figure 2 and [25]). Importantly, cells readily recovered a steady doubling time similar to exponential growth. This is in contrast with the growth arrest observed in *Escherichia coli* cells in response to various stresses, which results in long filamentous cells that eventually recover with shortened interdivision times [37]. Interestingly, exposure to DNA damaging agents and antibiotics can induce competence in several pathogens, including *S. pneumoniae* [17,38,39]. The use of a microfluidic device coupled with fluorescence microscopy, which has become an invaluable tool to study the physiology of bacterial cells in response to antibiotics in real-time measurements [40,41], now offers the possibility to further extend this field of research in the pneumococcus. In a broader perspective, this work provides new bases for the study, at the single cell level and in real time, of cellular processes that require optimal growth and long-term imaging of pneumococcal cells.

## Figures and Tables

**Figure 1 genes-11-00675-f001:**
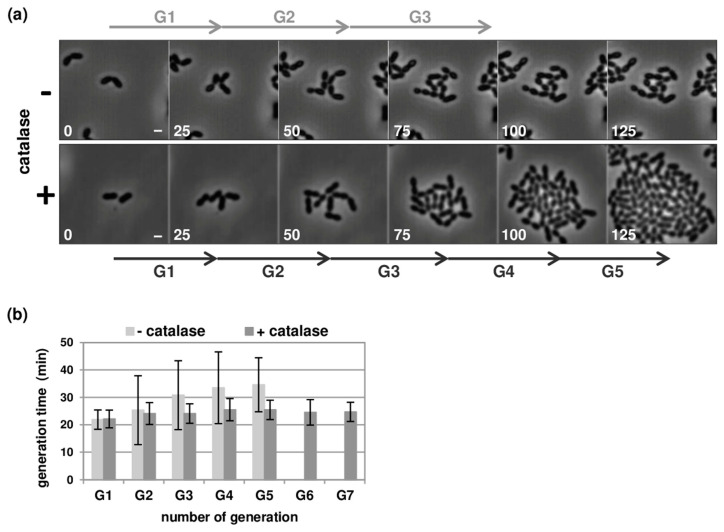
Growth of *Streptococcus pneumoniae* in the CellASIC^®^ ONIX microfluidic device. (**a**) Still images from time-lapse microscopy of R1501 (Δ*comC*) cells grown at 37 °C under a flow rates of 0.3 µL/h in C+Y medium supplemented (bottom panel) or not (top panel) with catalase (300 U/mL). Images were captured at 5-min intervals during 5 h. Representative phase contrast images captured during the two first hours are shown. Time is indicated in minutes. G1 to G5 and arrows indicate successive cell division events. Scale bar, 1 µm; (**b**) Histograms indicating the variation of the generation time over successive cell division events. Generation times were calculated by manually monitoring individual cells lineages over 7 rounds of division (G1 to G7). Note that cell growth stops after 5 cell division events without catalase. A minimum of 50 cell lineages were analyzed in each condition for each experiment. Values and standard deviations are based on data from three independent experiments.

**Figure 2 genes-11-00675-f002:**
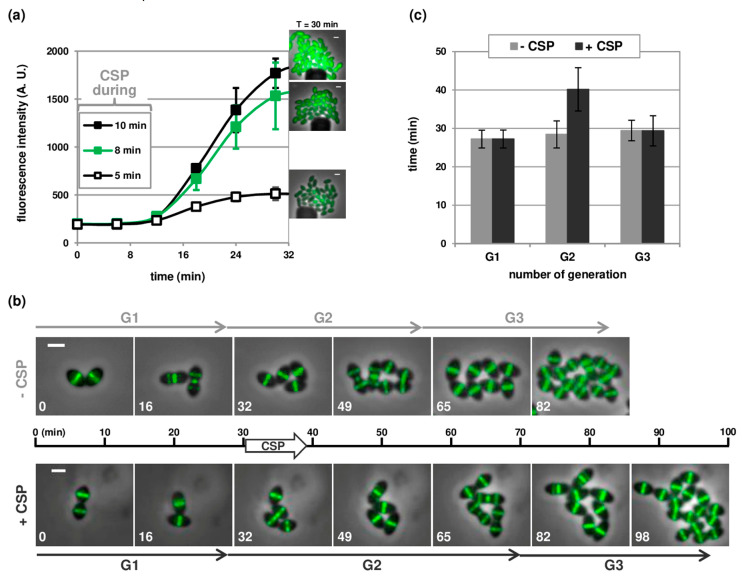
Competence-stimulating peptide (CSP)-dependent competence induction in microfluidic chambers. (**a**) Kinetics of expression of P_E_-*gfp* (strain R4254) in response to CSP addition at a concentration of 500 ng/µL and a flow rate of 23 µL/h (6 psi) for 5 min (open square), 8 min (green square) and 10 min (black square). Average fluorescence intensities per µm^2^ are based on time-lapse microscopy images and shown in arbitrary units (A.U.). More than 200 cells were analyzed per time point and per experiment. Time zero corresponds to the onset of CSP perfusion. Averages of three replicates are shown. Error bars show standard deviations. Some of the error bars are too small to be shown. Selected fluorescence microscopy images acquired at 30 min are shown on the right side. Overlays between phase contrast (gray) and Green Fluorescent Protein (GFP) (green) are shown. Scale bar, 1 µm; (**b**) Still images from fluorescence time-lapse microscopy of R3702 cells producing a functional FtsZ-GFP fusion. Cells were grown in two parallel microfluidic chambers and induced (+CSP) or not (−CSP) to develop competence by CSP perfusion 30 min after the beginning of the time-lapse acquisitions (see Section 2.3 for details). Overlays between phase contrast (grey) and GFP (green) are shown. Scale bar, 1 µm; (**c**) Histograms indicating the variation of the generation time of strain R3702 before, during and after CSP perfusion. Time-lapse microscopy images were captured at 5-min intervals. Generation times were calculated by manually monitoring individual cells lineages over 3 rounds of division (G1 to G3), with G1 corresponding to the generation before CSP perfusion, and G2 to the generation starting at the time of CSP perfusion. A minimum of 75 cell lineages were analyzed in each condition. Values and standard deviations are based on data from 2 independent experiments. Scale bars, 1 µm (note that the magnification is not identical with and without CSP).

**Figure 3 genes-11-00675-f003:**
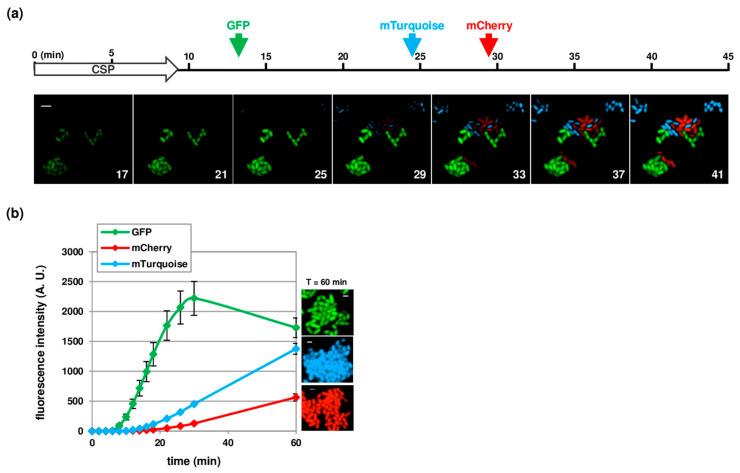
Comparison of the kinetics of expression of P_E_-*gfp* (strain R4254), P_E_-*mTurquoise* (strain R4255) and P_E_-*mCherry* (strain R4256) in response to CSP addition. (**a**) Still images from time-lapse microscopy of a mixed culture containing equal densities of all 3 strains. Images were captured at 2min intervals during 1 h. The starting point of the time-lapse corresponds to the beginning of CSP perfusion. First image shown corresponds to the 17th min. Every second image is shown. Time is given in minutes. Arrows indicate the initial time at which the fluorescence signal is visually detected for GFP (green), mTurquoise (blue) and mCherry (red). Overlays between GFP (green), mTurquoise (blue) and mCherry (red) are shown. Scale bar, 2 µm; (**b**) Quantification analysis of fluorescence signal based on time-lapse microscopy images. Strains R4254, R4255 and R4256 were grown in parallel microfluidic chambers and induced to develop competence by CSP perfusion. Time zero corresponds to the onset of CSP perfusion. Images were captured at 2 min intervals. Average fluorescence intensities per µm^2^ are shown in arbitrary units (A.U.). More than 200 cells were analyzed per time point and per experiment. Lines and confidence bands represent means of three replicates and standard deviations, respectively. GFP (green), mTurquoise (blue) and mCherry (red) fluorescence images acquired at 60 min are shown on the right side. Scale bar, 1 µm.

**Figure 4 genes-11-00675-f004:**
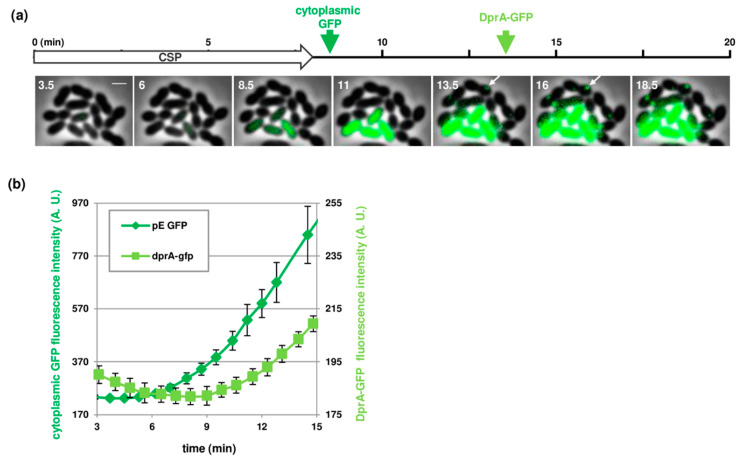
Comparison of the kinetics of expression of early (P_E_-*gfp*, strain R4254) and late (*dprA-gfp*, strain R3728) competence genes. (**a**) Still images from time-lapse microscopy of a mixed culture containing equal densities of the two strains. Images were captured at 50 s intervals during 2 h. Every third image is shown. The starting point of the time-lapse corresponds to the beginning of CSP perfusion. Time is shown in minutes. Green Arrows indicate the initial time at which the fluorescence signal is visually detected for cytoplasmic GFP (dark green) and DprA-GFP foci (light green). Overlays between phase contrast (grey) and GFP (green) are shown. White arrows point at DprA-GFP foci; (**b**) Quantification analysis of fluorescence signal based on time-lapse microscopy images. Strains R4254 and R3728 were grown in parallel microfluidic chambers and induced to develop competence by CSP perfusion. Images were captured at 50 s intervals during 2 h. Time zero corresponds to the onset of CSP perfusion. Average fluorescence intensities per µm^2^ are shown in arbitrary units (A.U.). More than 500 cells were analyzed per time point and per experiment. Averages of three replicates are shown. Error bars show standard deviations.

**Figure 5 genes-11-00675-f005:**
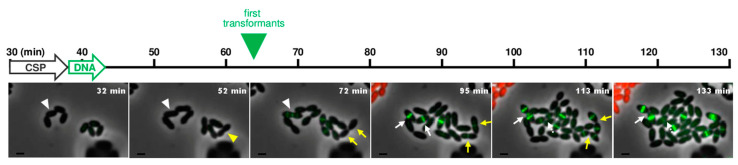
Near real-time visualization of transformation. Time-lapse microscopy of a mixed culture containing strain R3708 R3708 (*ftsZ*-stop-*gfp*) and strain R4256 (P_E_-*mCherry*). A time scale in minutes is shown. Images were captured at 4min intervals during 2.5 h. White arrow indicates the perfusion period of CSP (starting at 30 min after the beginning of the time-lapse), Green arrow indicates the perfusion period of donor DNA (PCR fragment encoding functional *ftsZ-gfp* fusion) and green triangle indicates the initial time at which the green FtsZ-GFP fluorescence signal (i.e., transformed cells) is visually detected. Overlays between phase contrast (grey), GFP (green) and mCherry (red) are shown. White arrowheads and arrow indicate transformed cells expressing FtsZ-GFP before the first cell division event, yellow arrowheads and arrows point at transformed cells expressing FtsZ-GFP after division. The data are representative of three biological replicates. Scale bar, 1 µm.

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
