# Peer review of "Direct Visualization of Horizontal Gene Transfer by Transformation in Live Pneumococcal Cells Using Microfluidics"

_genes, 2020, doi:10.3390/genes11060675_

Round 1

Reviewer 1 Report

The current manuscript by Mortier-Barriere et al., titled “Direct visualization of horizontal gene transfer by transformation in live pneumococcal cells using microfluidics” describes the development of a microfluidic method combined with fluorescence microscopy for time-lapse imaging of the competence development and transformation of the microaerophilic bacterium Streptococcus pneumoniae at the single-cell level. For this, the authors first optimized the growth conditions of the bacterium in microfluidic chambers to recreate the growth kinetics of the bacteria in batch liquid culture. Later, by generating transcriptional and translational fluorescent protein (gfp) reporter genes under the control of the promoters for the early or late com genes, the authors visualized the expression of early and late competent genes induced by addition of the competence-stimulating peptide (CSP). Finally, using two bacterial strains with PE-mCherry reporter gene and addition of ftsZ-gfp donor DNA fragment, the authors visualized the development of competence and transformation of the cells, respectively. The authors found that expression of the newly acquired genes continued for over 2 generations, supporting previous observations using traditional molecular biology based methods. Overall, this manuscript presents a microfluidics and fluorescence microscopy-based method to study bacterial transformation at the single-cell level. The method developed here has the potential to be used for studying other bacterial processes under various stimuli or stress conditions. The results in the manuscript support previous observations using classical studies; however, the manuscript falls short of novel and significant insights into bacterial transformation. The results are largely qualitative, since the analyses are subjective (visual analysis) and not very quantitative. The suggestions for further improving the manuscript, as a methods paper are outlined below:

Major changes

  1. Lines 17 and 23: The authors mention 'fluorescence microscopy to monitor pneumoccal competence development and transformation in real time and at the single cell level.', in the abstract and refer to the 'real time' imaging throughout the text. I think this is very misleading since the time-scale of the bacterial cell division (~24 min, here), and transformation is of the same order, as the time-scale of maturation of fluorescent proteins used in this study (especially mCherry, >20 min). I appreciate that the authors acknowledge this limitation of their imaging assay compared to earlier studies using Luciferase measurements (Lines 408-416 in Discussion). Therefore, I strongly suggest the authors to not refer to the imaging as 'real time', as it does not affect the results in the manuscript.
  2. Lines 264-272: 'Remarkably, however, the fluorescent signal was not detected….' Fluorescent proteins, especially the standard FPs used here have been thoroughly characterized and used for fluorescence microscopy for a long time. Therefore, there is nothing 'Remarkable' in the delayed observation of mTurquoise and especially mCherry in these studies. In addition, the results described in this whole paragraph and the associated Figure 3 are not novel, and therefore should be moved to the supplementary information. In the minimum, this paragraph can be just shortened quite a bit without the discussion on fluorescence extinction coefficient and quantum yield.
  3. Lines 272-274: 'Interestingly, we found that.….after complete shutoff.' – Please include a Supplementary Figure and/or a Supplementary Video to support this statement, as this is not obvious from the Figures.
  4. Figure 4a: Punctae/foci are hard to visualize in R3728 strain. These are clear in the Video S2, but not in the figure. The authors should consider making the figure panels larger.
  5. Result 3.4: What is the reason behind using strain R4256 with mCherry to visualize competence? Although it is nice to visualize both competence and transformation using the two strains, competence (mCherry signal) shows up much later than transformation (GFP fluorescence), which is reverse chronological. Why not instead use GFP for competence and mCherry for transformation?
  6. Following on the previous point, the authors could have used the right reporters, to get a rough estimate of the time between development of competence and the appearance of transformation, accounting for the maturation times of the two fluorophores (here GFP and mCherry). However, this would still only be an upper estimate for the time, since the assay is not exactly real-time. Such estimates of the times and kinetics will improve the quantitative nature and significance of the study. I still suggest the authors to include a value this time, in the results and/or discussion, along with citing any other study where this was measured.
  7. The authors' use of visual detection of fluorescence is very crude way of analyzing the data. This is because the time for 'visual detection' of florescence signal can vary for each fluorescent protein, depending on many factors such as the contrast settings, gain used for camera, etc. Therefore, it is more objective to define a minimal 'threshold' for the signal for a more thorough quantitative analysis of the kinetics. I suggest the authors to include the details of how they marked the times based on visual inspection in the Methods.

Minor changes

  1. I suggest using SI symbols for time – s and min instead of seconds and minutes
  2. Figure 2: Scale bar is not clearly visible in panel (a)– please show it in white and maybe on the top left corner in panel (b), to not be obscured by the time. Also, please add 'min' in the time bar (0-100 min) in panel (b).
  3. Please include Scale bar in Fig. 3 – the authors can add a longer 200 uM scale, for better visibility.
  4. Line 330: Please change 'consists in' to 'consists of' or 'based on'
  5. Figure 5: Change 'AND' to DNA.

Reviewer 2 Report

Mortier-Barrièreet al. describe how they develop the CellASCI ONiX microfluidicis platform for use to study pneumococcal competence and transformation and provide a framework for future studies using this microfluidics setup to study S. pneumoniae. The described methodology will be of interest to people in the pneumococcus field who wants to use microfluidics to study competence/transformation specifically but also more general for other single cell responses. The details provided in the manuscript regarding generation time and growth as a function of flow rates, addition of catalase etc are therefore very interesting. The manuscript is clearly written, and the methodology is mostly described with sufficient details. The data presented do not provide a lot of new insights into pneumococcal biology, but mainly confirms previously published data with this new setup.

I have a few specific points and questions which could be considered.

  1. Line 199-203 (and throughout). The authors use the term “injection” when adding CSP in the microfluidics system. I find this a bit confusing. As far as I understand “injection” in this context means to change the inlet to a well with CSP? Can the author rephrase or clarify this?
  2. Line 60-66. As the authors state, most of the information about pneumococcal competence and transformation comes from population data, and single cell studies with methods such as those presented in this manuscript would be highly valuable go get more insights into the process. However, I think the authors should also include information from single cell, real time, time-lapse data that are already available, including their own, Kurushima et al (2020) and other studies such as Berge et al. (2013), Moreno-Gamez et al. (2017), Kjos et al. (2016).
  3. Related to the previous point. In the discussion, the author should compare their results from the microfluidics setup with results obtained from time-lapse imaging of competence development and transformation without a microfluidics setup (ie on agar pads), including those mentioned above)
  4. 1 and Fig. S1. Fig. S1a (quantification of generation times) could be moved into the main manuscript as part of Fig. 1. I believe this plot provides important information that should be shown next to the micrographs.
  5. 2A. Is the competence development uniform in cells across the entire image frame – or is there any cell-to-cell variability with regard to timing or signal intensity?
  6. 2B. Please explain exactly how the FtsZ-signal was used as a marker for cell division. How do the authors know that the FtsZ-gfp fusion is functional, as stated in the legend? Any reference for that?
  7. 4. Is the DprA-GFP fusion (fully) functional? Could the use of this strain influence the competence and transformation dynamics compared to wild-type?
  8. 5 and line 341: It should be made clear that the R3708 contains the construct ftsZ-stop-gfp (stated in the legend, but should also be made clear in the text). Why did the authors not attempt to make a dual labelled strain, where both competence development and transformation could be visualized in the same strain?

Minor points:

  1. Line 108 (section about microfluidics methodology): were the cells always trapped in the same “height” of the microfluidics plates?
  2. Line 119 “…OD550 = 0.1.” Approximately how many cells/µl is this. Is this within the range of cell concentrations mentioned in the instructions for this microfluidics plate?
  3. Line 41. Consider replacing “and” with a comma: “…pathogenicity islands, antibiotic resistance …”
  4. 5. In the timeline ADN should be changed to DNA.
